# Mechanical Mechanism and Shaping Effect of Tunnel Blasting Construction in Rock with Weak Interlayer

**Mingfeng Lei** [1,2], **Rui He** [1,2], **Linghui Liu** [1,2,*], **Ningxin Sun** [3], **Guifang Qin** [4] and **Yunliang Zhang** [1,2]

1   School of Civil Engineering, Central South University, Changsha 410075, China
2   MOE Key Laboratory of Engineering Structures of Heavy Haul Railway, Central South University, Changsha 410075, China
3   Shandong Provincial Communications Planning and Design Institute Group Co., Ltd., Jinan 250031, China
4   Guizhou Road & Bridge Group Co., Ltd., Guiyang 550081, China
*   Correspondence: liulinghui888@163.com

**Abstract:** The weak interlayer, as a problematic geological body during tunnel construction, greatly influences the propagation of the blasting stress wave, the blasting excavation qualities, and the explosion efficiency. A series of numerical models were established to study the changes in the propagation process of blasting stress waves and the failure morphology of the surrounding rock mass, aiming to reveal the weak interlayer's influence mechanism. The result indicates that the weak interlayer's existence reduces the propagation velocity and stress peak of the stress wave at barred zones but strengthens the peak stress at reflection zones, which leads to an asymmetrical distribution of rock damage. Furthermore, the type and distribution of the weak interlayer were classified and generalized into four types. The tunnel blasting outlines under different types of weak interlayers are derived through numerical modeling for designing references. A strategy to resist tunnel overbreak and underbreak was proposed combined with previous work. The actual blasting solution is compared to the designed blasting solution with optimised blasting parameters.

**Keywords:** tunnel construction; barrier effect; blasting shaping effect; weak interlayer; stress wave propagation; overbreak and underbreak

## 1. Introduction

The weak interlayer, commonly formed by geological deposition or ground movement, exists in all types of rock masses. The low strength and stiffness of the interlayer greatly influence tunnel stability. Also, the weak interlayer in the rock mass affects the propagation of the blasting stress wave and the formation of blasting cracks, thereby changing the fracture mode of the rock mass. The hard-control blasting construction process easily causes over-excavation, under-excavation, and even tunnel collapse [1–5]. Thus, the propagation characteristic of stress waves and the blasting mechanism become critical issues in drilling and blasting tunnel design and construction.

The weak interlayer seriously impacts the propagation and attenuation of stress waves on the structure surface; many scholars have conducted theoretical analysis on the aspect [6–13]. For instance, Haskell [14] used the transfer matrix method to study stress waves' refraction and reflection characteristics penetrating multilayered geological media. Kause et al. [15] established a stiffness matrix method for solving stress wave propagation in multi-layered geological media. Xu et al. [16] proposed an analytical solution for the propagation of plane harmonics in an elastic interlayer. They discussed the relationship of blasting parameters reflection coefficient, transmission coefficient, interlayer thickness, and incident angle. Fan et al. [17] utilized plane P waves to analyze the stress wave's reflection energy coefficient and transmission energy coefficient propagating through the sandwich. Perino et al. [18] used the displacement discontinuity method and the equivalent medium method to analyze the propagation of stress waves in rock masses with weak

interlayers and conducted a comparison analysis. Hu et al. [19] used the series solution of the Legendre equation to explore the waveform changing law through the weak interlayer and the damage result. Lei et al. [20] proposed an improved equivalent viscoelastic medium method to study the influence of stress wave incident angle and interlayer thickness on stress wave propagation. Briefly, the transmission and reflection law of the stress wave on the structure surface has been widely analyzed. However, few studies focus on the vibration pattern and attenuation law of the wave vibration frequency as the stress wave passes through the weak interlayer.

Additionally, the weak interlayer impacts the rock mass's stability [21,22]. Zheng et al. [23] analyzed the excavation stability of rock caverns under weak interlayer conditions. Huang et al. [24] carried out a physical model test to compare the failure modes of a homogeneous formation and a weak interlayer surrounding rock and believed that the weak interlayer caused asymmetric stress distribution and affected the stability of the tunnel. Panthee et al. [25] believe that structures such as weak interlayers control the size of tunnel blasting over- and under-excavation. Zhao et al. [26] summarized the crack development process during blasting and excavation of rock masses with weak interlayers through field micro-seismic (MS) monitoring methods. Zhang et al. [27] obtained the rock mass deformation parameter formulas of soft and hard layered rock masses under different stress directions. Lv et al. [28] studied the failure mode and constitutive model parameters of weak interbedded rock mass. Liu et al. [29] proposed that the inclination angle of the weak interlayer is the most sensitive parameter that affects the deformation of the rock mass. The sensitivity order of other parameters lists as follows: thickness, elastic modulus, cohesion, and internal friction angle. However, most existing research studied the static stability influence of weak interlayers on the rock mass. Still, it rarely concerned the dynamic influence characteristics of the weak interlayer and the final blasting profile forming effect. In contrast, the excavation outline formed by drilling and blasting is the following construction basis.

Aiming to reveal the stress wave propagation mechanism, relevant numerical simulations and field tests were carried out in this paper. Then, the classification and generalization of the weak interlayer's spatial distribution and shape were carried out. Finally, a conceptual model with a different impact region is proposed, and the blasting outline of weakly interbedded tunnels with other geometrical parameters are derived for design reference. To summarize, the technology roadmap is plotted in Figure 1.

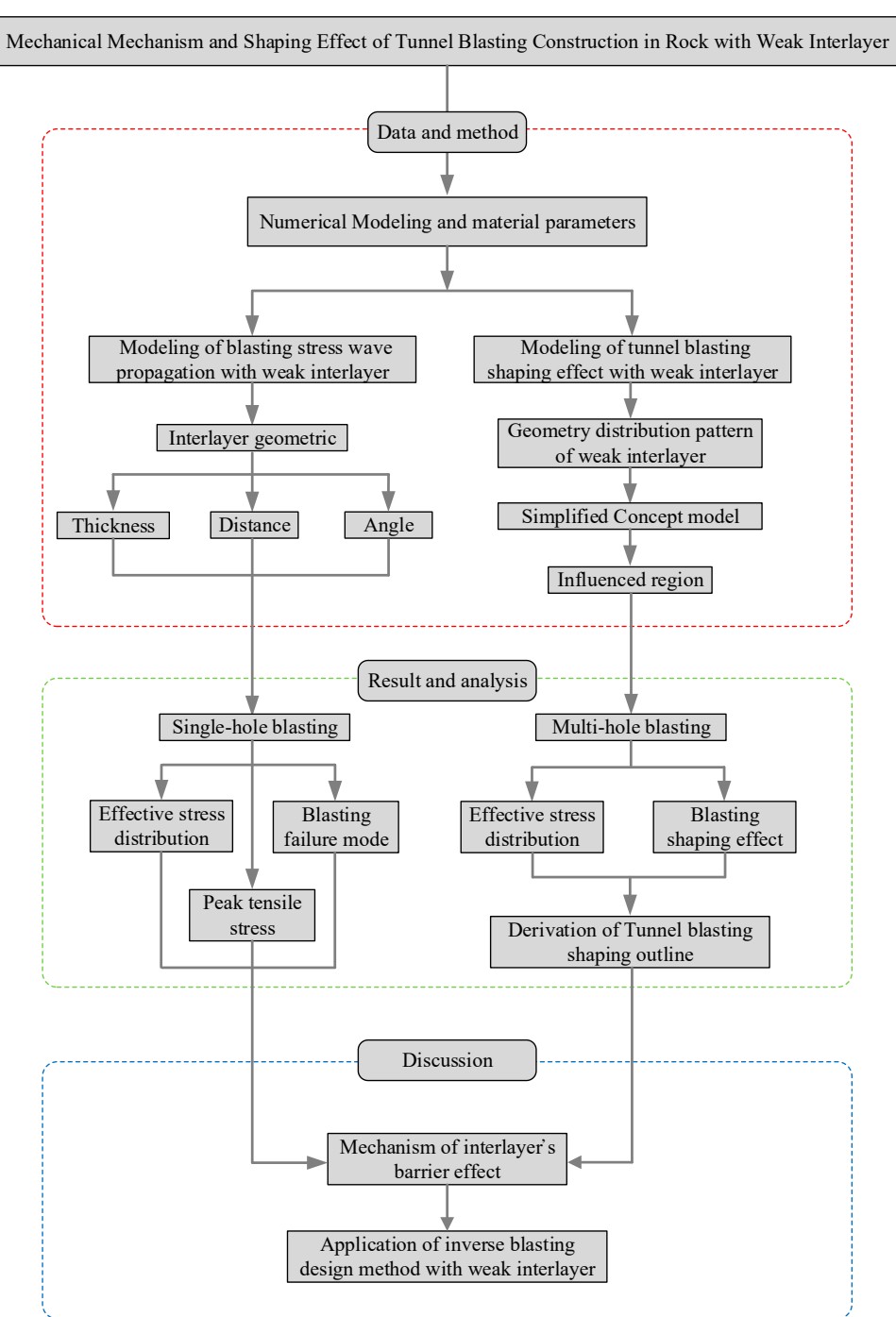

**Figure 1.** Technology roadmap.

## 2. Data and Method

### 2.1. Numerical Modeling and Material Parameters

To simulate the dynamical blasting process and the stress wave propagation, the dynamic explicit program ANSYS/LS-DYNA was applied. The element **solid 164** is selected for the simulation of rock, interlayer, and explosive. The constitutive model of bilinear kinematic hardening with strain-rate independence is adopted for rock deformation analysis, where the strain rate is calculated by the Cowper–Symonds model [30]. Using the Von Mises criterion, the rock stress field distribution and blasting fracture could be simulated by defining the key field *MAT_ADD_EROSION [31]: grid cells "quit" as blasting

stress exceeds its dynamic tensile stress. The rock's dynamic yield stress $\sigma_y$ and strain rate $\dot{\varepsilon}$ satisfy the following relationship:

$$\sigma_y = \left[1 + \left(\frac{\dot{\varepsilon}}{C}\right)^{\frac{1}{P}}\right]\left(\sigma_0 + \beta E_P \varepsilon_P{}^{eff}\right) \tag{1}$$

where $\sigma_0$ is the initial yield stress, $\dot{\varepsilon}$ is the strain rate, $C$ and $P$ are constants related to material properties, $\varepsilon_P{}^{eff}$ is the effective plastic strain, and $E_P$ is the plastic hardening modulus $E_P = E_{\text{tan}}E/(E - E_{\text{tan}})$.

The research area typically has sandstone rock mass embedded carbonaceous slate weak interlayers. Yang et al. [32] proposed the stress–strain relationship under different strain rates of the rock mass. According to his study and combined with the research results of Xia et al. [33], the two strain rate parameters $C$ and $P$ are 2.63 and 3.96, respectively. Table 1 shows the material parameters of the surrounding rocks and weak interlayer. Wang points out that the rock mass's dynamic tensile strength is 1-10 times its static tensile strength [34]; thus, the dynamic tensile strength of the sandstone is considered to reach up to 35 MPa.

**Table 1.** Mechanical parameters of rock and weak interlayer.

| Parameters | Sandstone | Carbonaceous Slate |
|---|---|---|
| Density $\rho$/kg·m$^{-3}$ | 2600 | 2200 |
| Elastic modulus $E$/GPa | 37.5 | 10 |
| Poisson ratio $\mu$ | 0.27 | 0.31 |
| Tensile Strength $\sigma_t$/MPa | 5 | 2 |
| Plastic modulus $E_P$/GPa | 0.0375 | 0.01 |
| Hardening parameters $\beta$ | 0.6 | 0.6 |
| Strain rate parameter $C$/S$^{-1}$ | 2.63 | 2.63 |
| Strain rate parameter $P$ | 3.96 | 3.96 |

The explosive chosen corresponds to the emulsion type that is commonly used in engineering. Its physical and mechanical parameters are shown in Table 2. The keyword *MAT_HIGH_EXPLOSIVE_BURN is set to simulate the explosion process, and the JWL equation is used to define the stress and volumetric strain as follows:

$$P = A(1 - \frac{w}{R_1 V})e^{-R_1 V} + B(1 - \frac{w}{R_2 V}) + \frac{wE_0}{V} \tag{2}$$

where $P$ is the blasting-produced pressure; $V$ is the relative volume; $E_0$ is the initial specific energy; and $A$, $B$, $R_1$, $R_2$, and $w$ are constants.

The explosive is arranged in the blast hole as a non-coupling charge structure with the non-coupling medium air. In the LS-DYNA program, the stress variation is simulated using the linear polynomial state equation *EOS_LINEAR_POLYNOMIAL:

$$P = C_0 + C_1\mu + C_2\mu^2 + (C_3\mu^3 + C_4 + C_5\mu + C_6\mu^2)E_0 \tag{3}$$

where $C_0$ through $C_6$ are constants, the other symbols are the same as in Equations (1) and (2), and the corresponding parameters are shown in Table 3.

**Table 2.** Material parameters of No. 2 rock emulsion explosive.

| $P$/kg·m$^{-3}$ | $D$/m·s$^{-1}$ | $A$/GPa | $B$/GPa | $R_1$ | $R_2$ | $\omega$ | $E_0$/GPa |
|---|---|---|---|---|---|---|---|
| 1000 | 3400 | 229 | 0.55 | 6.5 | 1.0 | 0.35 | 3.51 |

**Table 3.** Material parameters of air.

| $\rho$/kg·m$^{-3}$ | $C_0$ | $C_1$ | $C_2$ | $C_3$ | $C_4$ | $C_5$ | $C_6$ | $E_0$/GPa |
|---|---|---|---|---|---|---|---|---|
| 1.29 | 0 | 0 | 0 | 0 | 0.4 | 0.4 | 0 | 0.025 |

Additionally, to overcome the non-convergence problem, some modeling techniques were used. For example, the simulation of rock adopts the Lagrange algorithm, while the explosive and air are defined using the ALE [35] algorithm. The fluid-structure interaction is set between the rock, the air, and the explosive.

### 2.2. Modeling of Blasting Stress Wave Propagation with Weak Interlayer

In the propagation analysis, the geometrical characteristics of the weak interlayer can be characterized by the interlayer thickness ($h$), the horizontal distance from the hole center to the weak interlayer central line ($d$), and the angle between the weak interlayer central line and the line $l$ ($\theta$), as shown in Figure 2. The single blasting hole, with a 42 mm diameter and 16 mm cartridge diameter, is placed at the model center. An uncoupling coefficient of 2.63 and center detonation are set in the model. Figure 3 shows the size of the blast model ($h$ = 10 cm, $d$ = 25 cm, $\theta$ = 90°).

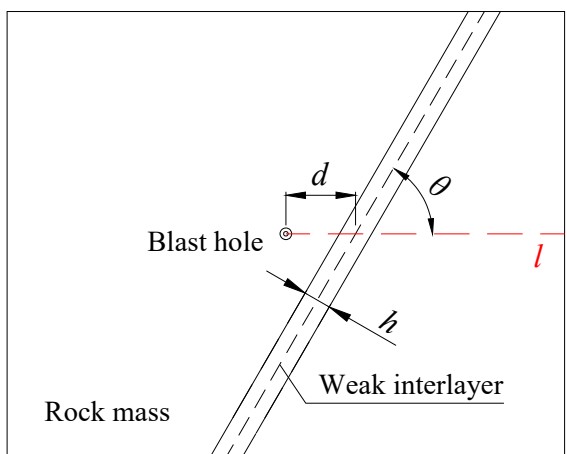

**Figure 2.** Geometric position of the weak interlayer and blast hole.

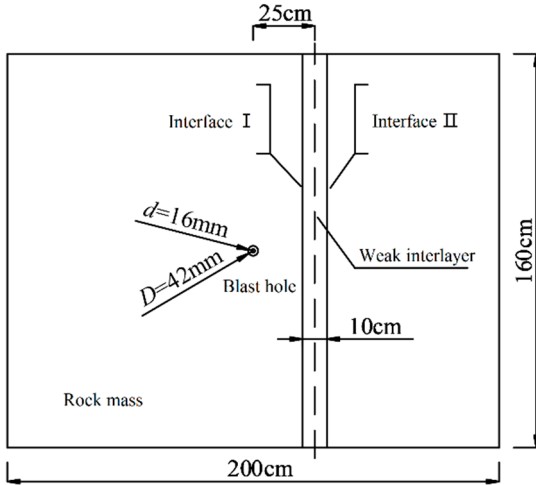

**Figure 3.** Blasting model of rock mass with weak interlayer.

A thin plate mesh grid conducted the calculation simplification. Considering the symmetry and small size of the blast hole, the quasi-2D problem was simplified through plate modeling and mesh mapping. The mesh size could then be controlled within 1 cm, and the

mesh was refined at the blast hole. Finally, the whole model is divided into 64,448 elements and 130,340 nodes and uses a normal constraint and non-reflective boundary condition. The final mesh grid is shown in Figure 4.

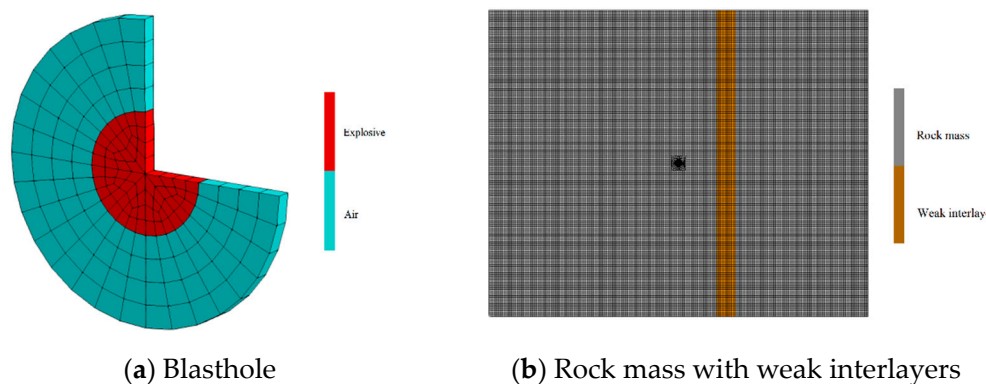

(**a**) Blasthole　　　　　　　　　　　　(**b**) Rock mass with weak interlayers

**Figure 4.** Mesh grid of rock mass and weak interlayer.

Then, the calculations were divided into three groups to adjust and analyze the influence of thickness, distance, and angle of the interlayer. Nine measurement points were selected on the line to observe the stress response around the interlayer. Table 4 lists the detailed arrangement of each single-hole condition. (Note that the thickness group considers $d = 0$ cm as the single-hole condition V.)

**Table 4.** Calculation conditions and arrangement of stress measuring points.

| Factors | Arrangement of Measuring Points | Single-Hole Conditions | | | |
|---|---|---|---|---|---|
| | | I | II | III | IV |
| Thickness $h$/cm | 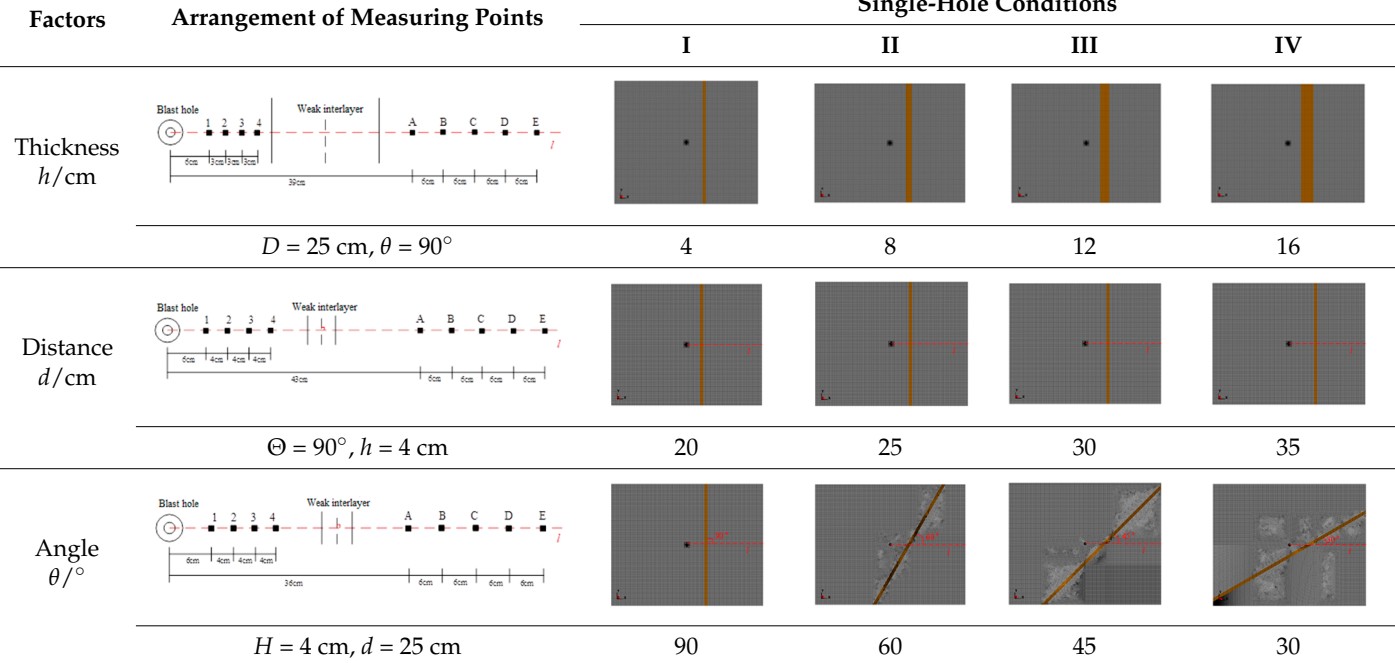 | | | | |
| | $D = 25$ cm, $\theta = 90°$ | 4 | 8 | 12 | 16 |
| Distance $d$/cm | | | | | |
| | $\Theta = 90°$, $h = 4$ cm | 20 | 25 | 30 | 35 |
| Angle $\theta$/° | | | | | |
| | $H = 4$ cm, $d = 25$ cm | 90 | 60 | 45 | 30 |

### 2.3. Modeling of Tunnel Blasting Shaping Effect with Weak Interlayer

In practice, blasting projects involve numerous blasting holes, such as the borehole-blasting method in tunnel construction. Hence, the blasting shaping research shall expand the single-hole condition to multiple holes. The smooth blasting method generally has a good blasting shaping effect aiming at the tunnel outline. The rational design of the blast hole arrangement and the explosive charge of the method could control the overbreak or underbreak excavation, and further protect the surrounding rock. However, the blasting design requires generalized parameters to characterize the complex spatial distribution of

weak interlayers. Aiming at the blasting design, the following simplification restrictions are proposed for the blasting shaping simulation:

(1) Ignoring the interlayer's longitudinal distribution and spatial impact, the blasting shaping effect research is regarded as a 2D problem, and only the tunnel cross-section is considered.

(2) The studied weak interlayer is a thin and soft rock material embedded in the intact and hard rock mass, with a thickness from a few centimeters to tens of centimeters.

Determining the position relationship of the weak interlayer and the blasting outline, the spatial distribution of the interlayer can be summarized, as shown in Figure 5.

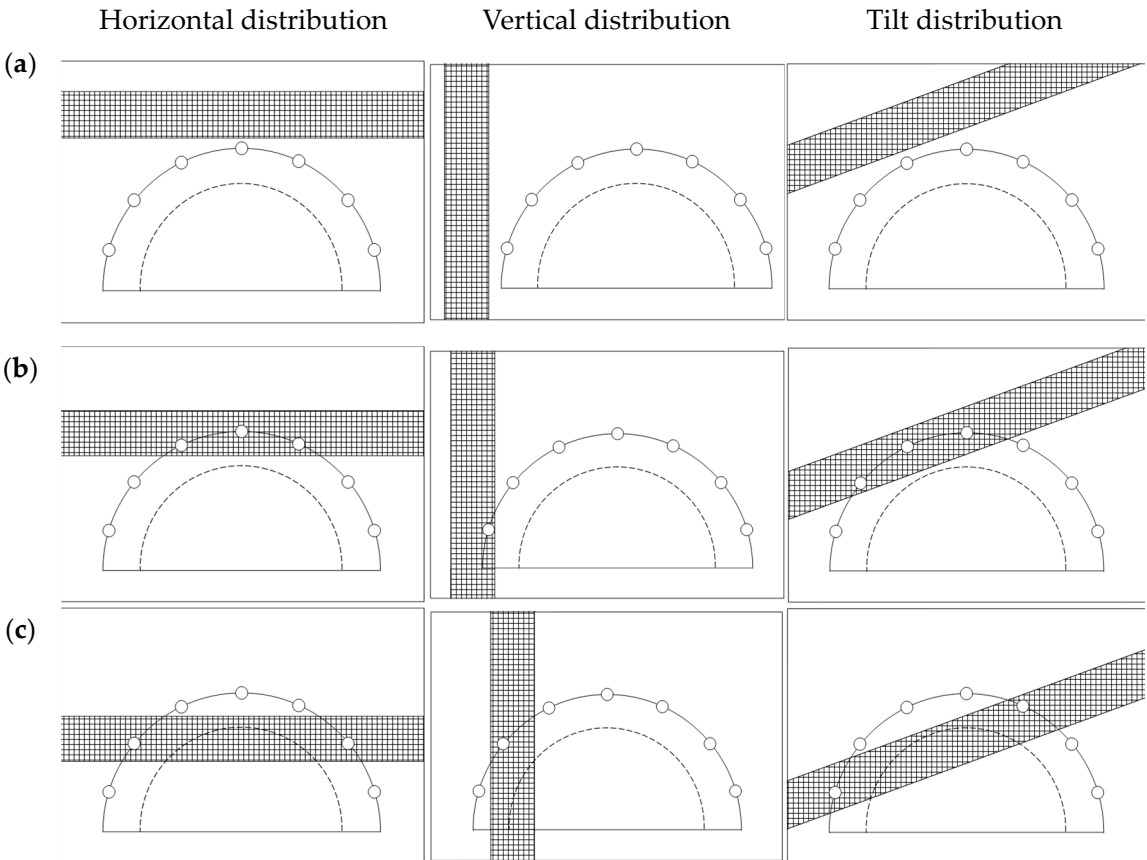

**Figure 5.** Geometry distribution pattern of weak interlayer: (**a**) outside the smooth blasting layer; (**b**) partially intersected with the smooth blasting layer; (**c**) intersected with the smooth blasting layer.

The numerous conditions of the interlayer distribution pattern may suffer from the complex influence and may still lead to confusing conclusions. Thus, several simplified conceptual models were established to analyze the blasting effect with a thin interlayer as follows:

(1) Figure 6 shows the approximate blasting influenced region of the thin interlayer (red dashed box). The only affected holes are those near the interlayer (15 times the borehole diameter). In Figure 6a, the blast holes in region A are affected, whereas others are not influenced.

(2) Compared with the influence of the blasting stress wave, the influence of gravity is relatively small. Thus, regions A, B, and C in Figure 6 can be regarded as equivalent regions since the relationship of the interlayer and the blast layer has no difference, and the only differences are the position and angle.

Therefore, a simplified model called the "influenced region" was proposed based on the above analysis: a blast-affected region unrelated to the spatial location distribution, as shown in Figure 6d, the region is simulated.

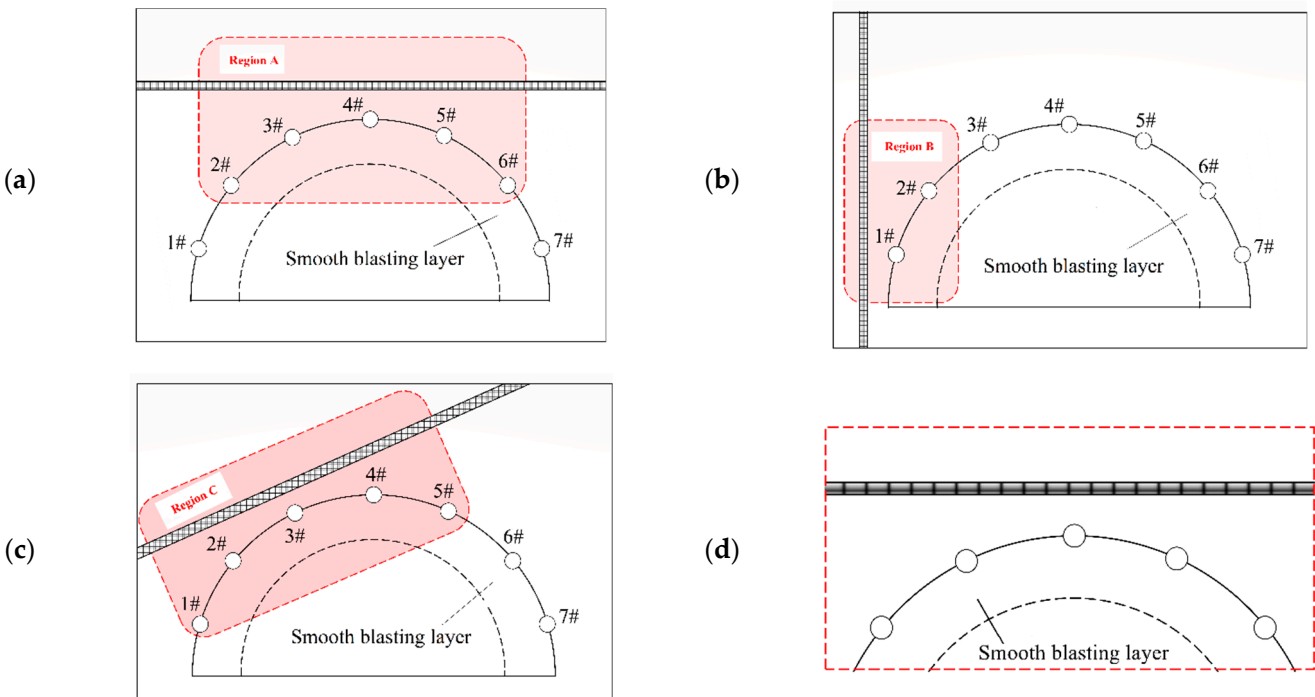

**Figure 6.** Influenced region as interlayer outside tunnel excavation outline: (**a**) Horizontal distribution; (**b**) Vertical distribution; (**c**) Tilt distribution; (**d**) Modeling region.

Figure 5. The conditions share the same parameters except for the thickness and spatial distribution of the weak interlayer in order to analyze the geometrical influence.

Figure 7 shows the model details of working condition 2. The thickness of the weak interlayer is 50 cm with an inclination angle of 0 °. Thirteen blasting holes have a diameter of 42 mm and spacing of 50 cm. The explosive diameter is 16 mm with a radial decoupling coefficient of 2.63. The thickness of the smooth blasting layer is 60 cm under the assumption that the excavation meets the design requirement. The model's overall size is 700 cm × 300 cm, and non-reflective boundary conditions are added to simulate the infinity of surrounding rock. Additionally, the blue dashed part is the blasting's free surface that adds non-reflective boundary conditions (Table 5).

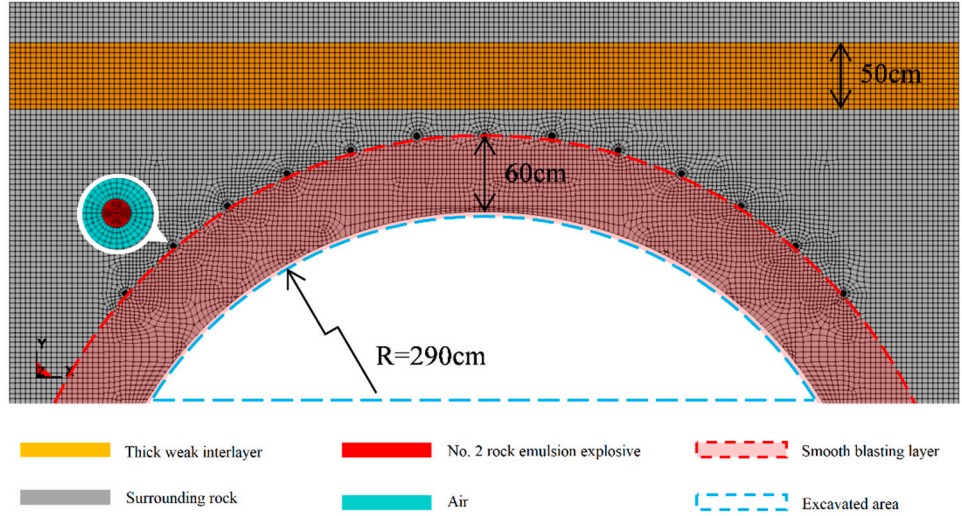

**Figure 7.** Finite element model of influenced region (condition 2).

**Table 5.** Calculation conditions.

| Multi-Hole Condition | Geometric Position | Concept Model | Numerical Model |
|---|---|---|---|
| I | No weak interlayer exists |  |  |
| II | Thick interlayer outside the tunnel excavation outline |  |  |
| III | Thick interlayer intersects with the inner contour of smooth blasting layer |  |  |
| IV | Thin interlayer intersects with the outer contour of smooth blasting layer |  |  |

## 3. Result and Analysis

### 3.1. Blasting Stress Wave Propagation

Figures 8–10 shows the effective blasting stress field at $t$ = 130 μs, the peak stress distribution, and the rock blasting failure effect. Some general performance can be concluded: in an intact and homogeneous rock mass with no weak interlayer, the stress wave displays a cylindrical pattern with a spacing distribution of compression and sparse waves. The overall intensity of the stress wave decreases along with the propagation distance. However, when the weak interlayer exists, it may induce stress transmission and reflection at the interface between weak and intact rock and redistribution of the stress wave, called the barrier effect. Commonly, the geometry and location of the interlayer, such as the interlayer thickness, the distance, and the angle, both impact the effect. The rock mass's effective stress distribution, peak stress, and blasting failure mode are analyzed separately from the three groups.

#### 3.1.1. Effective Stress Distribution

With the increase of the weak interlayer thickness, the barrier effect is enhanced, the propagation velocity of the stress wave decreases, and the peak time of the stress wave behind the interlayer delays. Meanwhile, the intensity attenuation velocity accelerates as peak stress reduces. The differences are demonstrated in Figure 8. In contrast, the stress wave presents an elliptic pattern when passing through the interlayer, as shown in Figure 9. A small distance between the hole and the weak interlayer can reflect huge converged stress energy that significantly enhances the interlayer's stress intensity. However, as the included angle $\theta$ reduces, the vertical distance from the initiation center to the weak interlayer decreases. The decreasing distance enhances the effective stress and peak stress

near the blasting hole. Thus, stress wave energy primarily converges in the normal direction of the weak interlayer (Figure 10).

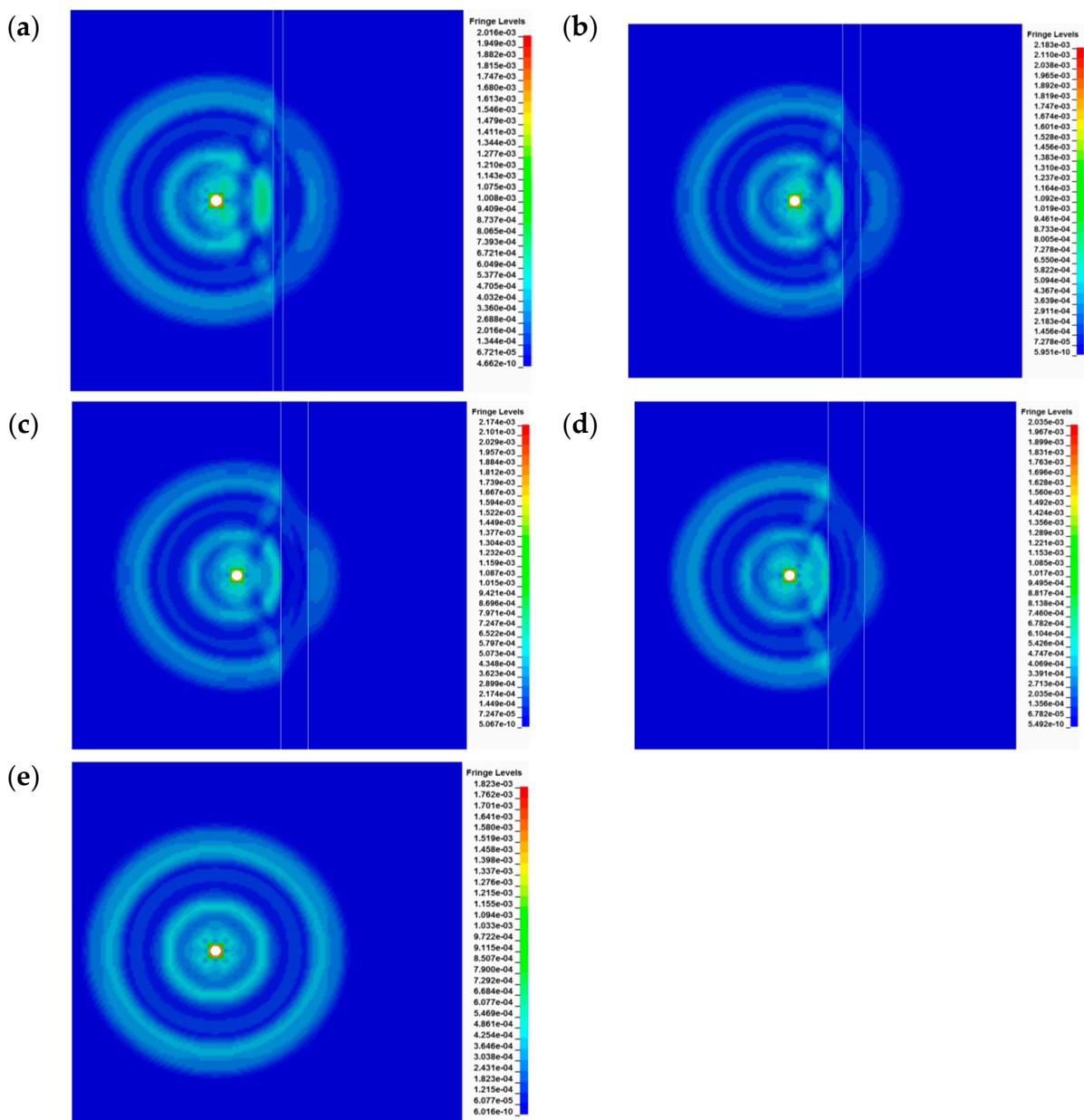

**Figure 8.** Effective stress distribution under different thickness: (**a**) single-hole condition I; (**b**) single-hole II; (**c**) single-hole condition III; (**d**) single-hole condition IV; (**e**) single-hole condition V.

### 3.1.2. Peak Tensile Stress

Figure 11 shows the relationship between the peak tensile stress and the propagation distance of the blasting wave. As shown in Figure 11a, the peak stress decreased continuously with the stress wave propagation in an intact rock mass. The reflection effect induces tensile stress waves in the reflection region, increasing peak tensile stress, and the increment becomes remarkable when approaching the weak interlayer. As the interlayer thickness increases, the intensity of the reflected tensile stress is strengthened, and the peak tensile stress at the measuring point grows with an increasing growth rate. In Figure 11b, the effective stress decreases linearly with the propagation distance in the intact rock mass. Meanwhile, the reflection and transmission effect of the weak interlayer induces stress-energy loss and the intensity reduction of stress waves behind the interlayer. The

thickness of the weak interlayer *h* and average stress attenuation rate fits with a parabolic relationship, as shown in Figure 12.

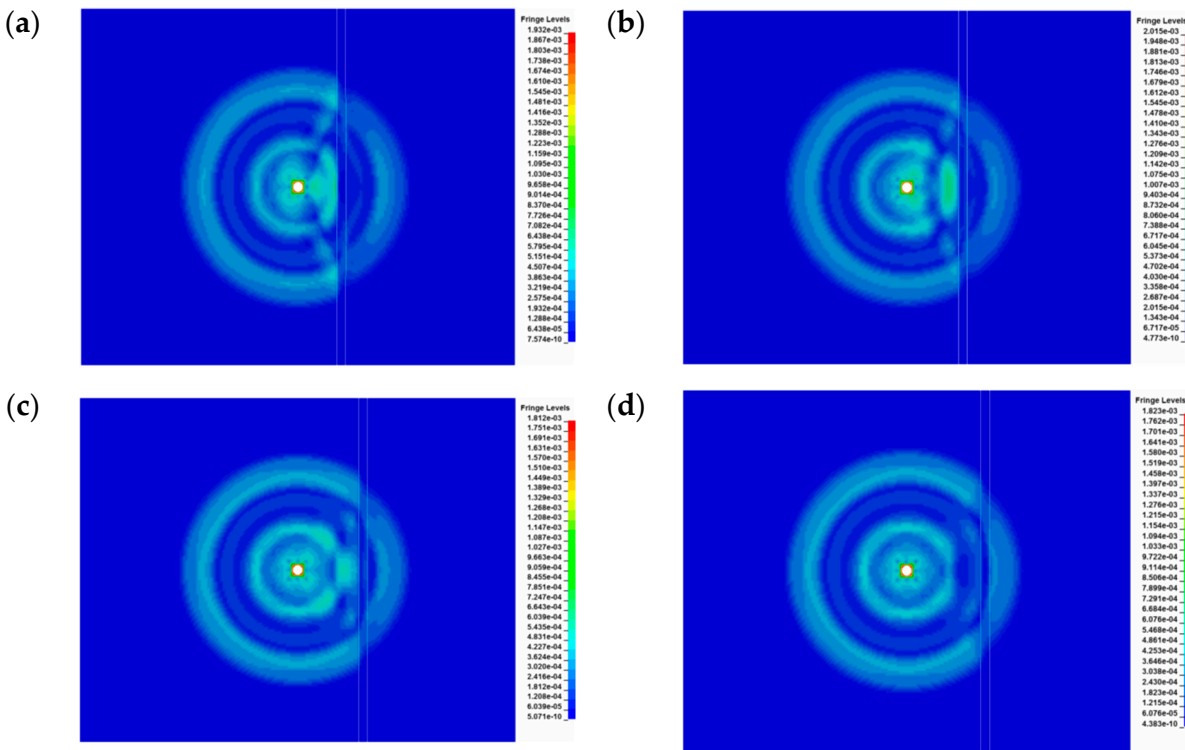

**Figure 9.** Effective stress distribution under different geometric positions: (**a**) single-hole condition I; (**b**) single-hole II; (**c**) single-hole condition III; (**d**) single-hole condition IV.

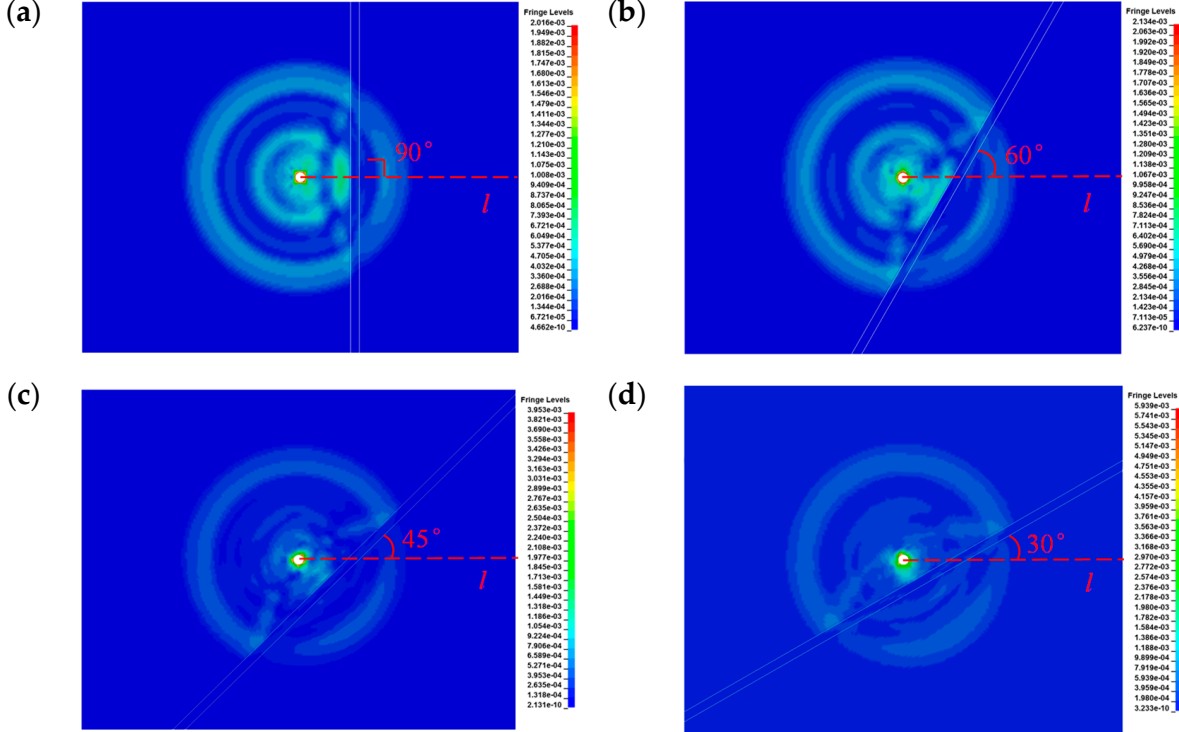

**Figure 10.** Effective stress distribution under different angle: (**a**) single-hole condition I; (**b**) single-hole II; (**c**) single-hole condition III; (**d**) single-hole condition IV.

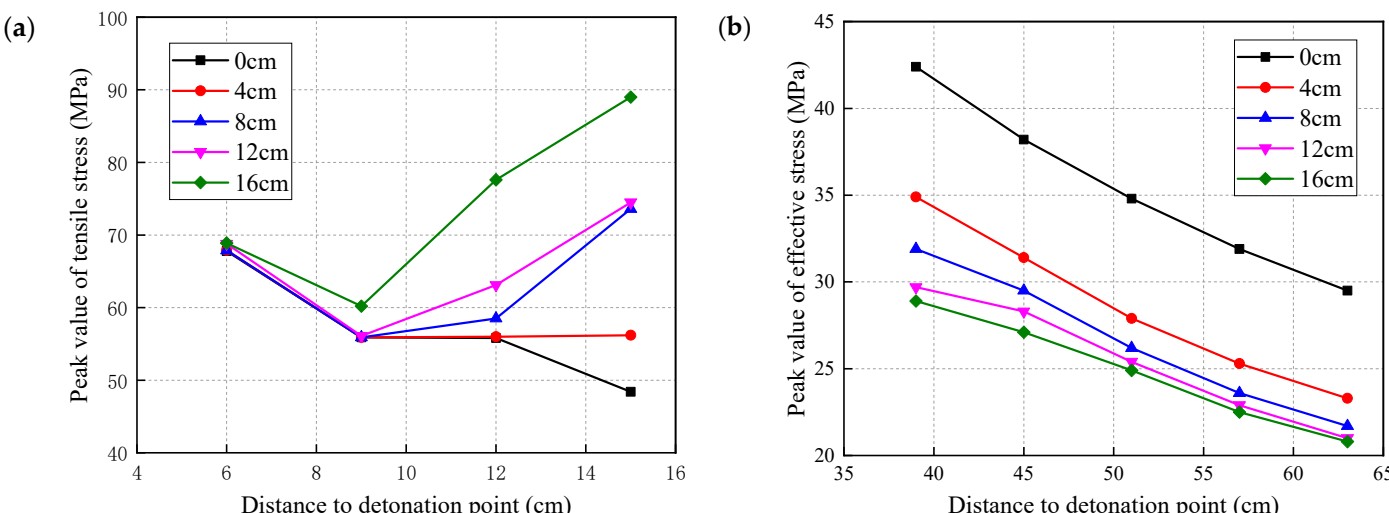

**Figure 11.** Stress peak curve varies with interlayer thickness under single-hole conditions: (**a**) Tensile stress at #1-4 measuring points; (**b**) Effective stress at measuring point A-E.

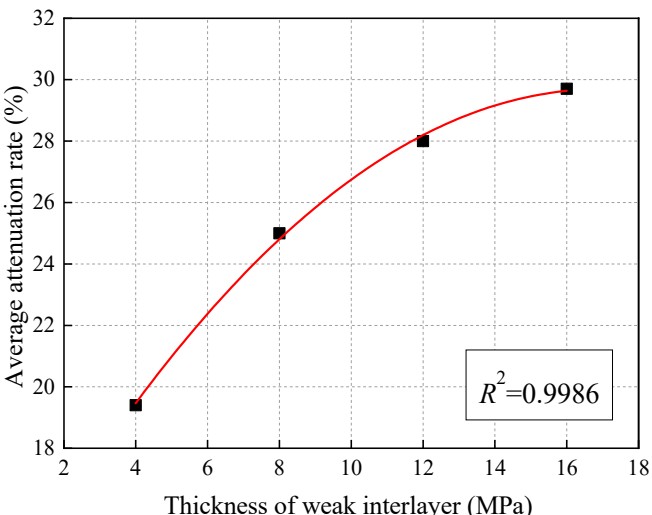

**Figure 12.** Average attenuation rate of effective stress peak.

With the distance increase between the blasting hole and the interlayer, the increasing rate of the peak stress and the promotion effect gradually move from measuring point #1 to #4, as shown in Figure 13. The distance increase promotes the peak tensile stress decrease at measuring point #1, decreases the peak stress at measuring point #1, but gradually reduces the peak stress differences at measuring points #1, #2, #3, and #4. That indicates that the stress wave's influence gradually diminishes as the stress wave propagates, and the peak stress increases due to the decrease of the vertical distance.

As to the included angle, a small included angle induces a small vertical distance from the initiation center to the weak interlayer interface I, as shown in Figure 14a. At the measuring points in the normal direction of the interlayer, the peak tensile stress is attenuated slowly, and the attenuation rate is inversely proportional to the angle. Similarly, the energy of the stress wave is primarily concentrated in the normal direction of the weak interlayer. As shown in Figure 14b, a smaller interlayer included angle can produce a greater barrier effect on the stress wave. Moreover, with the increase of the propagation distance, the influence of the angle variation on the stress wave propagation is gradually weakened. Additionally, the effective stress and peak tensile stress are enhanced around the blasting hole, the peak tensile stress at measuring point #1 increases, and the stress

waves in the direction of the measuring line *l* are converted from normal incidence to oblique incidence.

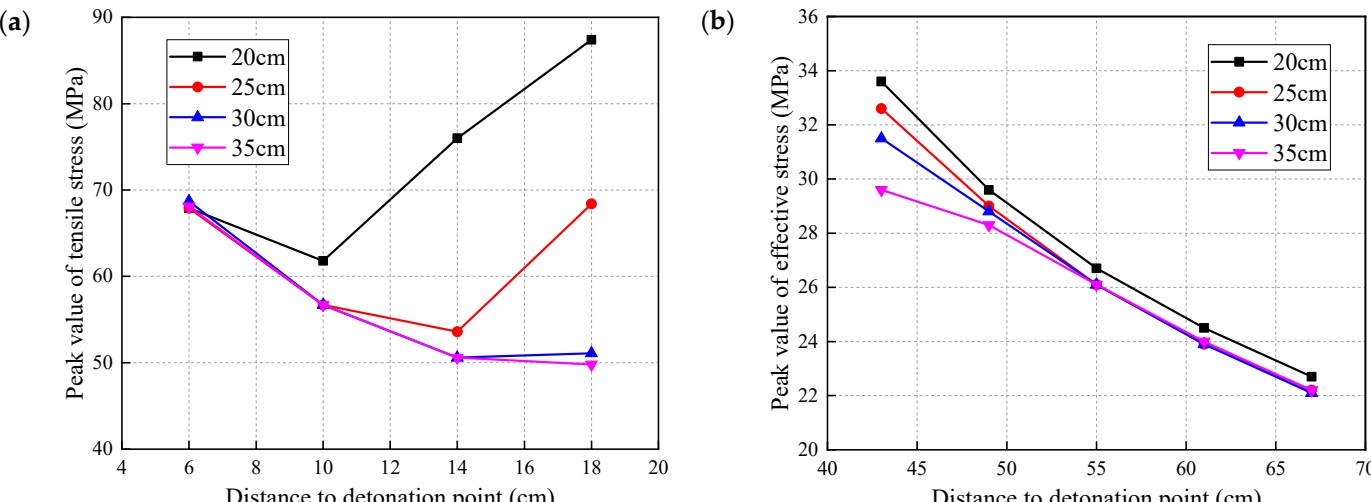

**Figure 13.** Stress peak curve varies with interlayer geometric position under single-hole conditions: (**a**) Tensile stress at #1-4 measuring points; (**b**) Effective stress at measuring point A-E.

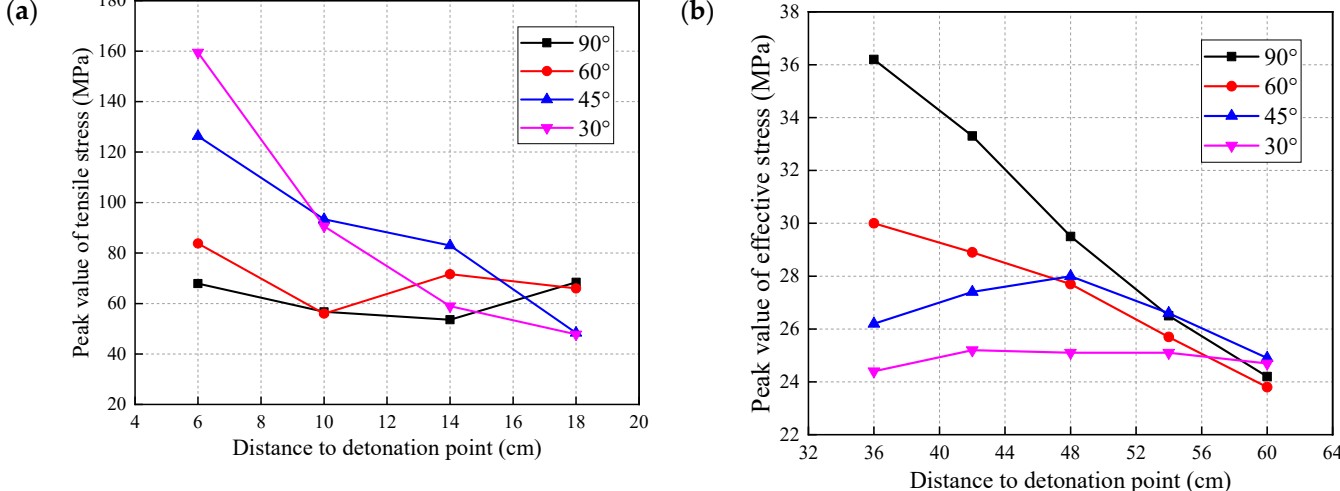

**Figure 14.** Stress peak curve varies with angles under single-hole conditions: (**a**) Tensile stress at #1-4 measuring points; (**b**) Effective stress at measuring point A-E.

### 3.1.3. The Blasting Failure Mode

The rock blasting failure mode is shown in Figures 15–17. Blasting cracks propagate freely and show a symmetry failure pattern in an intact rock mass. The weak interlayer converges the blasting energy on the front, enhancing the region's rock damage. Additionally, the intrusion of the blasting gas inserts the interlayer, breaking the interlayer and causing energy loss. The interlayer forms a blocking barrier to resist the damage propagation. A thinner interlayer may expand the brokerage; a thick interlayer displays a converged broken pattern.

Meanwhile, a large distance represents sparse blasting cracks ahead of the weak interlayer, whereas a small distance can induce dense blasting cracks. The energy converges at the interlayer front significantly raise the local principal tensile stress, leading to the exacerbated damage of rock mass.

The blasting effect of the rock mass corresponding to the four working conditions is shown in Figure 15. The vertical direction of the weak interlayer is mostly beneficial for

stress wave reflection. Moreover, the blasting center is closest to the weak interlayer in the vertical direction, corresponding to the highest stress wave intensity. Thus, the blasting cracks primarily develop along the direction perpendicular to the weak interlayer. The barrier effect of the interlayer on the stress wave and the penetrating effect of the blasting gas cause energy losses. Regarding measuring line *l* as the concentric line of the blast hole, the weak interlayer will significantly influence the shaping effect of the blasting tunnel hole. When the included angle *θ* becomes smaller, the blasting crack deviates from the direction of the blasting centerline and expands towards the direction normal to the weak interlayer.

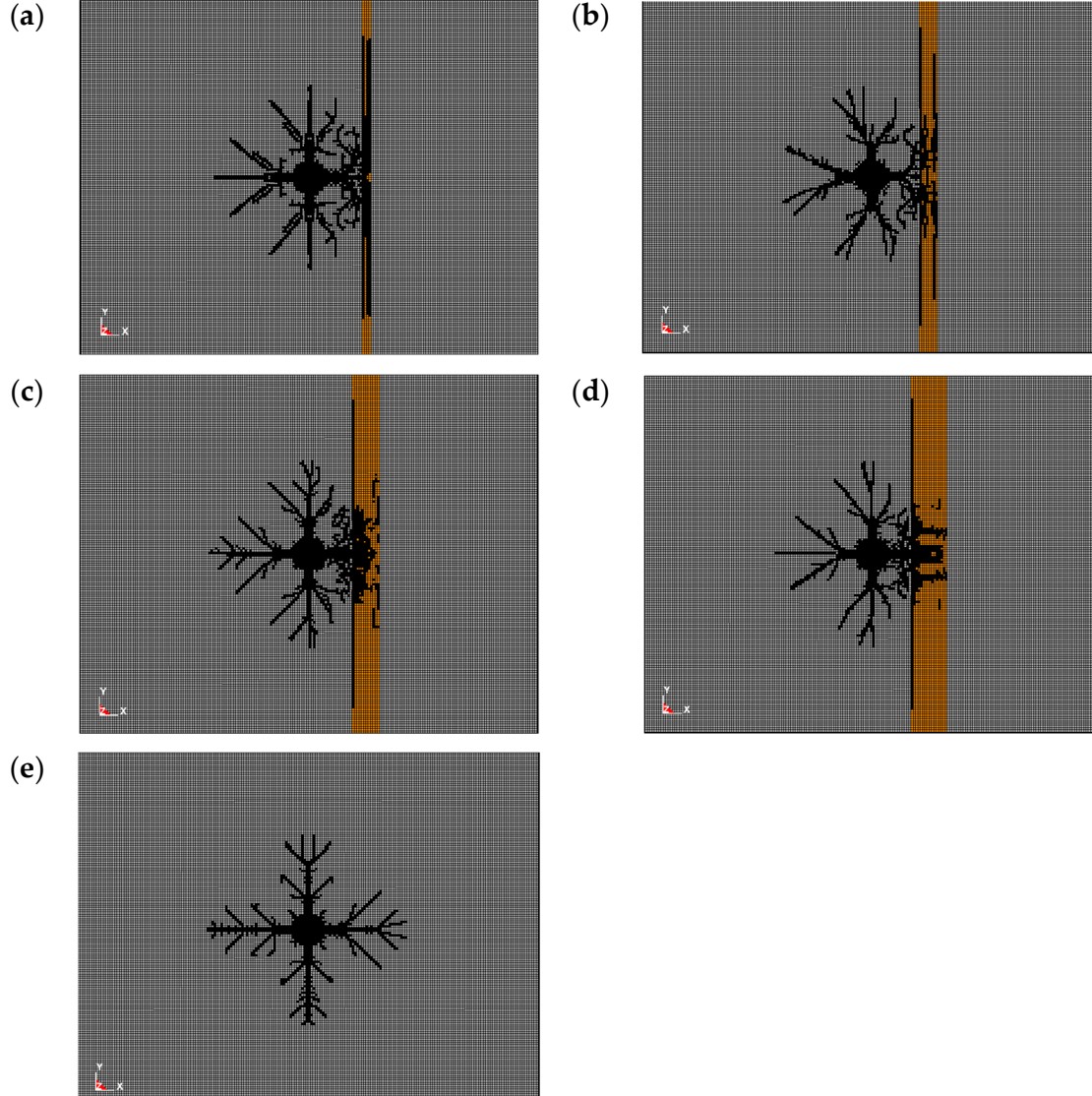

**Figure 15.** Blasting failure mode under different interlayer thicknesses: (**a**) single-hole condition I; (**b**) single-hole condition II; (**c**) single-hole condition III; (**d**) single-hole condition IV; (**e**) single-hole condition V.

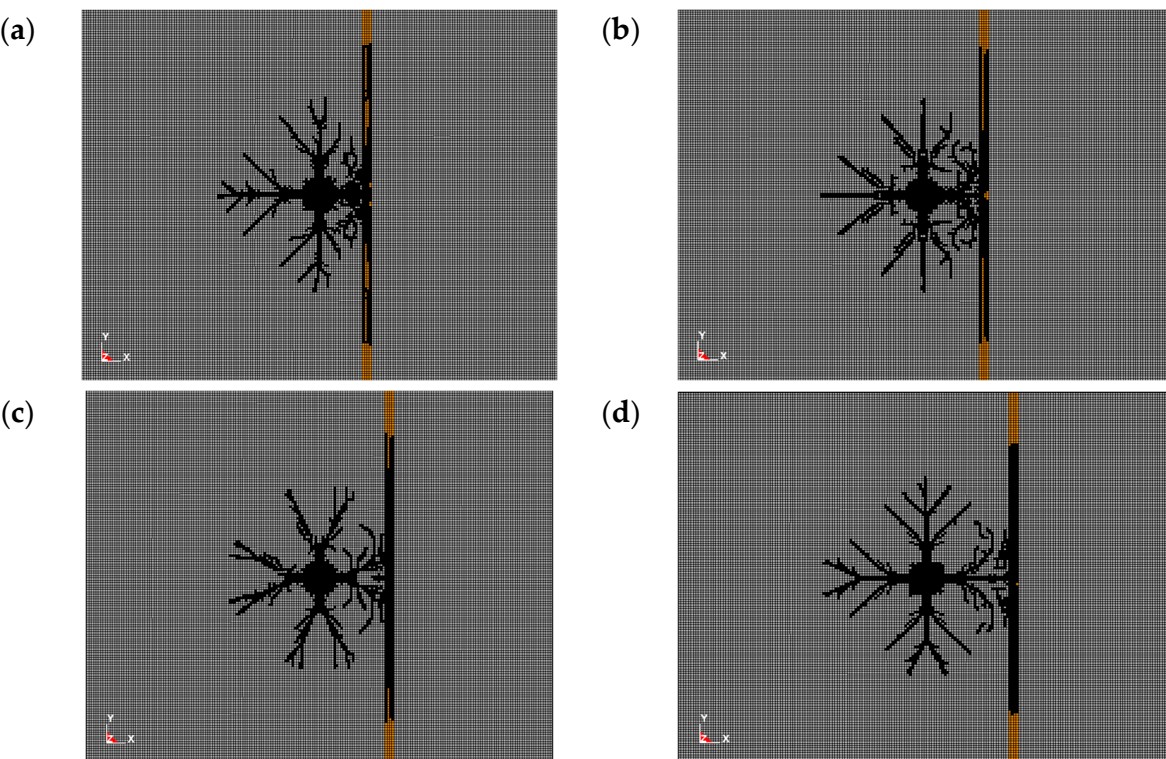

**Figure 16.** Blasting failure mode under different interlayer positions: (**a**) single-hole condition I; (**b**) single-hole condition II; (**c**) single-hole condition III; (**d**) single-hole condition IV.

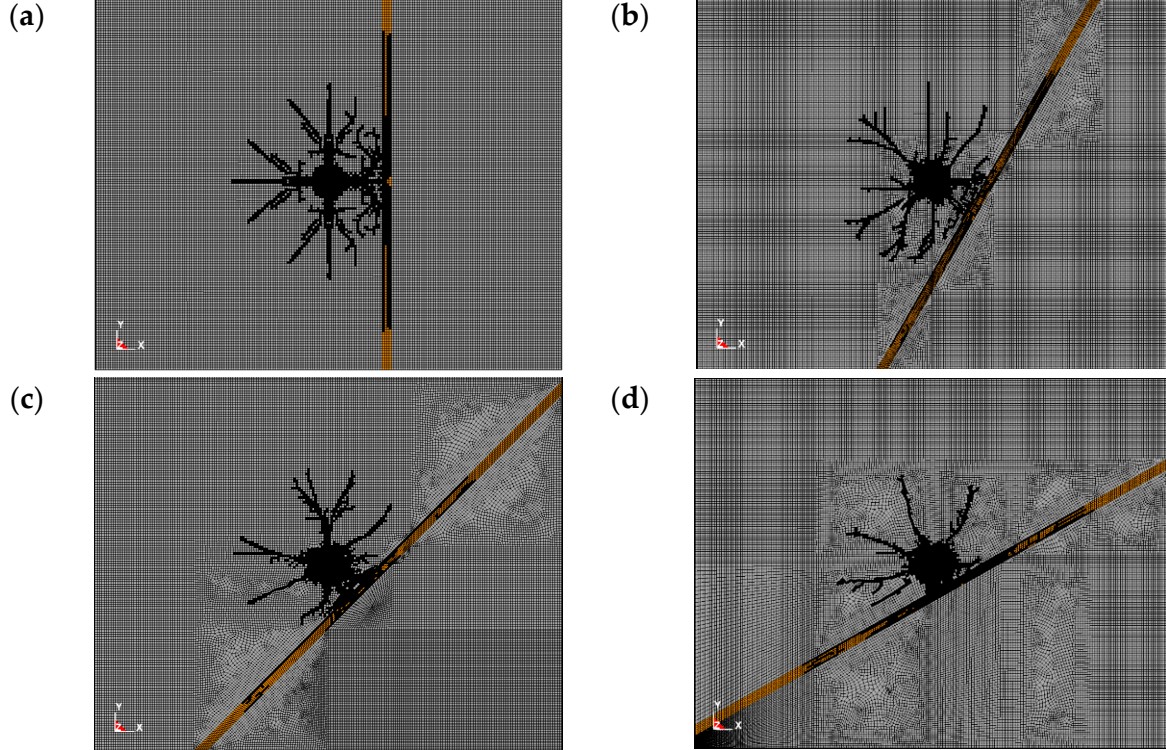

**Figure 17.** Blasting damage mode under different interlayer angles: (**a**) single-hole condition I; (**b**) single-hole II; (**c**) single-hole condition III; (**d**) single-hole condition IV.

### 3.2. Tunnel Blasting Shaping Effect

Figure 18 shows the distribution of the effective stress field and the final blasting effect at t = 150 μs after blasting. A regular excavation blasting face with superior shaping effect can be obtained (condition I). The result indicates that: the intact rock mass can obtain a regular excavation blasting face with a superior shaping effect. However, intensified rock damage occurs in the vault region with a thick weak interlayer (condition II). The overbreak vault is formed with a poor blasting effect. When the weak interlayer inserts the tunnel outline, blasting causes severer damage rather than the adjacent rock due to the low intensity of the weak interlayer, resulting in overbreak at the hance region (condition III) and poor blasting effect. In contrast, the thin interlayer hinders the propagation and superposition of the stress wave. Additionally, under the penetration by the blasting gas, the rock of the interlayer first undergoes bedding failure, and the insufficient blasting also leads to the preservation of large amounts of blocks in the vault region in smooth blasting layers (condition IV).

| Multi-hole Condition | Effective stress distribution (t = 150 μs) | Blasting shaping effect |
|:---:|:---:|:---:|

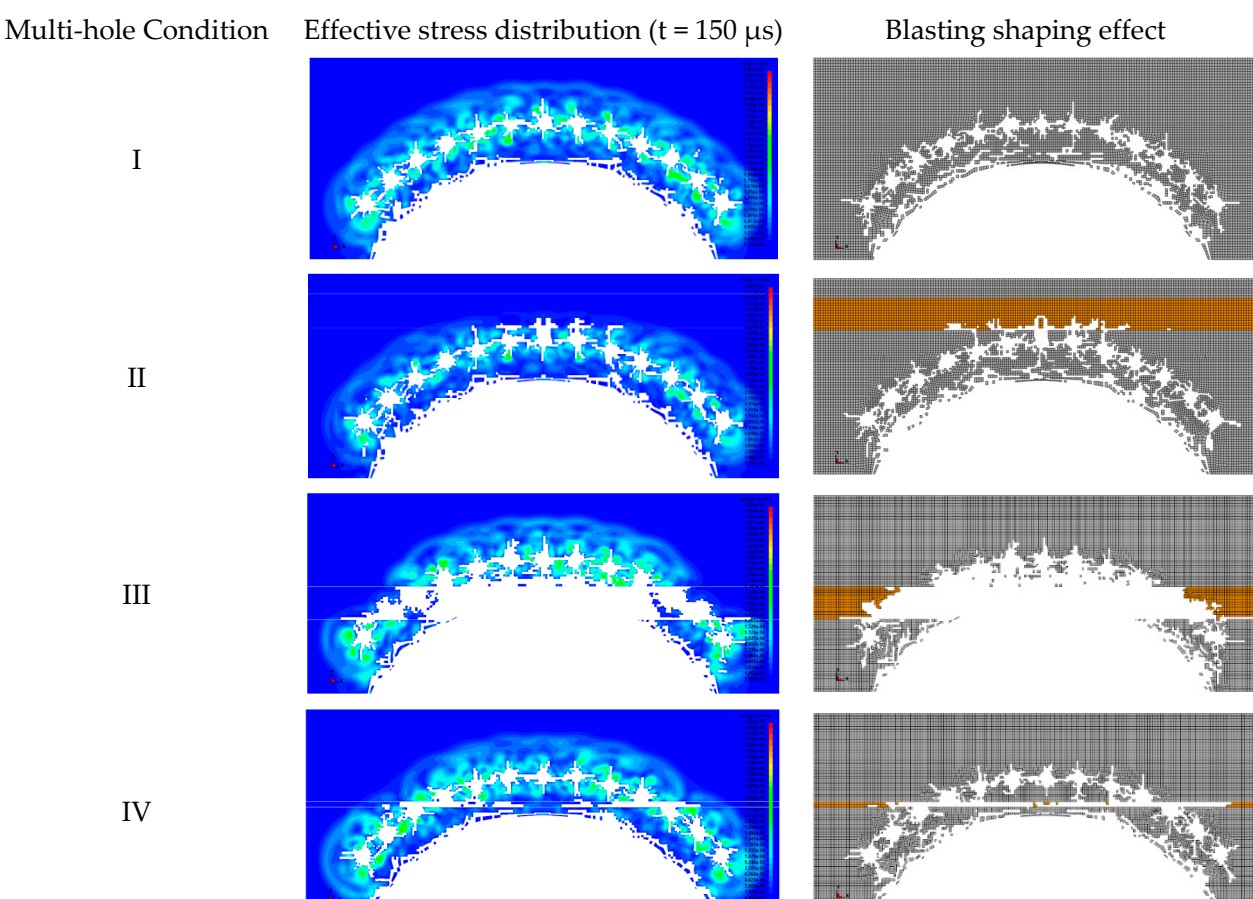

**Figure 18.** Effective stress distribution and final blasting shaping effect diagram of the influenced region in each condition.

Apparently, the existence of the thin and thick weak interlayer seriously affects the smooth blasting effect. Aiming at the practice, the blasting design scheme needs to be specifically revised with the inverse analysis of the blasting shaping results.

### 3.3. Devation of the Tunnel Blasting Shaping Outline

Referring to the simulated result, the blasting shaping effect with other interlayer geometrical distributions can be quantificationally drawn. Specifically, the drawing method is first to calculate the relative position and geometry of the interlayer and then deduce the corresponding blasting shaping outline utilizing the geometrical relation. The final

results with thin and thick interlayers are shown in Figures 19 and 20. The solid red line is regarded as the tunnel blasting outline, and the black dotted line is the contour line of the smooth blasting layer and the weak interlayer. The inverse result can provide significant reference for smooth blasting design.

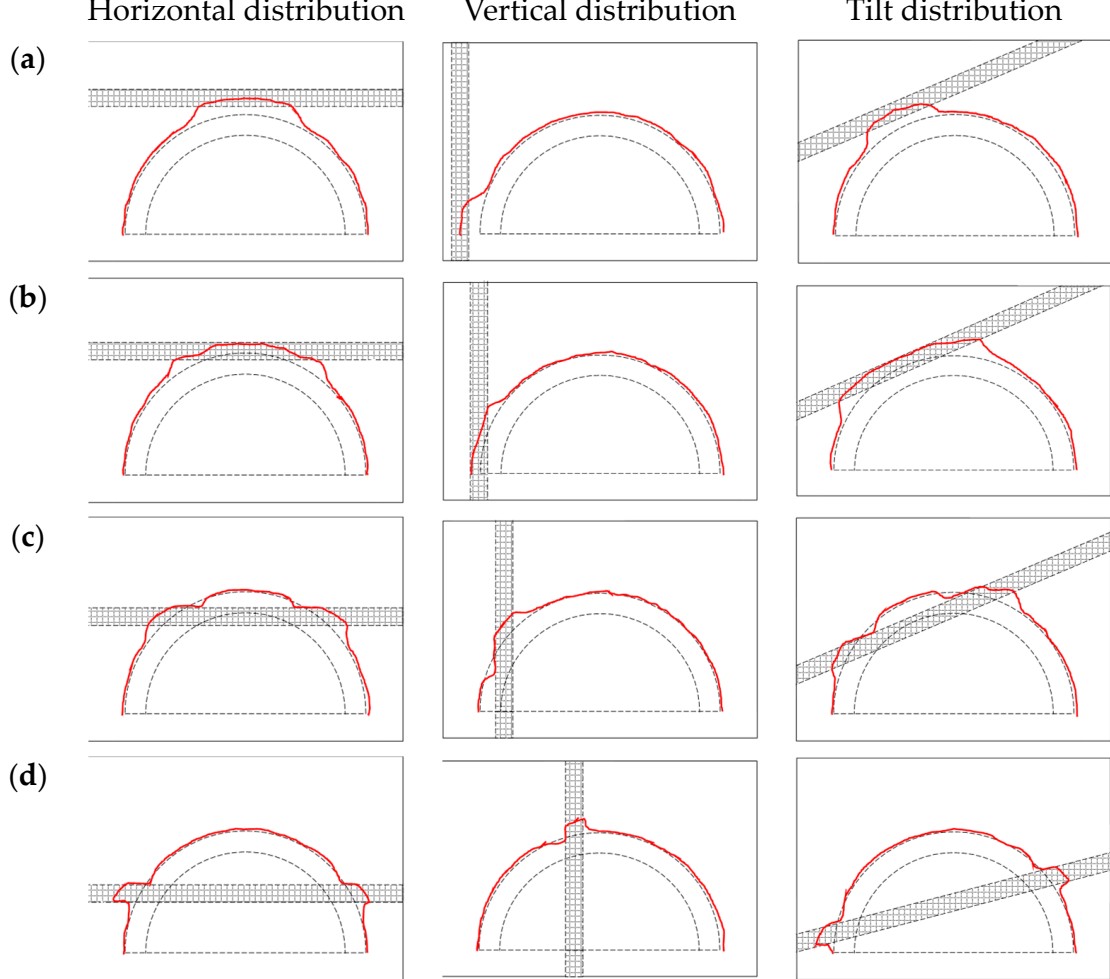

**Figure 19.** Blasting shaping outlines vary with different positions of thick interlayer: (**a**) outside the excavation contour; (**b**) intersects with the outer contour line of the smooth blasting layer; (**c**) intersects with the inner contour of the smooth blasting layer; (**d**) completely intersects with the inner contour of the smooth blasting layer.

### 3.4. Application of the Tunnel Blasting Shaping Outline

To demonstrate the function and effectiveness of the blasting shaping outline in tunnel blasting design, the practical application is demonstrated in separated double-line tunnel construction in China. The length of the left part is 3024 m, and the right is 3049 m. The strata in the tunnel construction region are composed of the Diluvial proluvial accumulation layer and the Triassic weathered sandstone. Due to insufficient attention to the weak interlayer around the tunnel, visible overbreak phenomenons were observed at the primary construction stage. A typical shaping outline in the overbreak section is shown in Figure 21. Similarly, the overbreak outline geometry is close to the deduced results, which verifies the accuracy of the previous derivation.

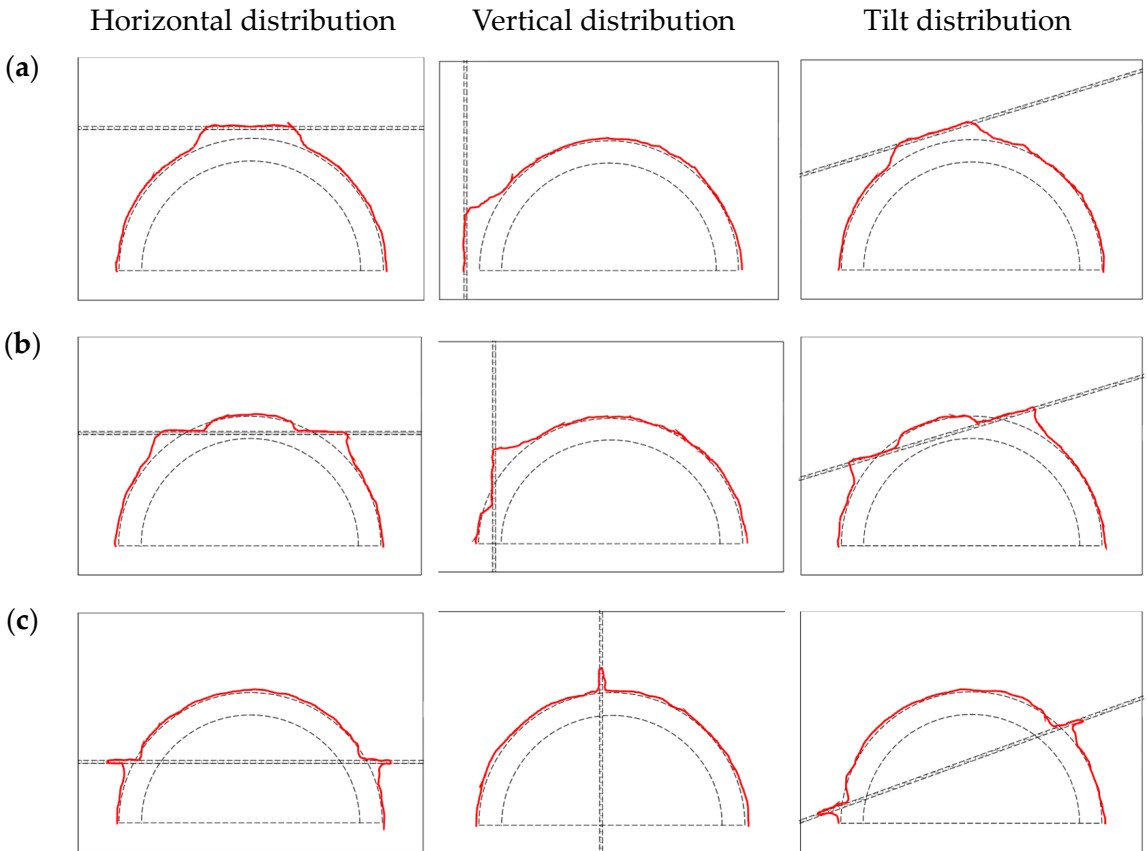

**Figure 20.** Blasting shaping outlines vary with different position of thin interlayer: (**a**) outside the excavation contour; (**b**) intersects with the outer contour line of the smooth blasting layer; (**c**) intersects with the inner contour of the smooth blasting layer.

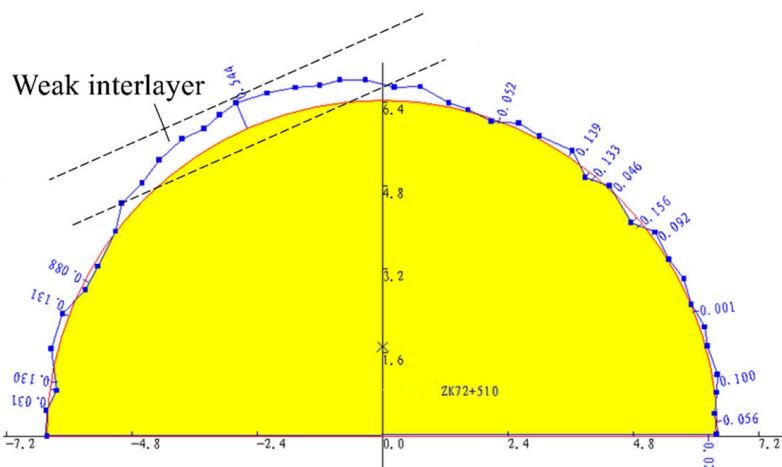

**Figure 21.** Tunnel blasting effect with insufficient consideration of the weak interlayer.

### 3.4.1. Overbreak Mechanism

According to numerical results, the overbreak region can be divided into two types referring to crack forming mechanism, the regions A and B, as shown in Figure 22. The weak interlayers are comprised of carbonaceous slate that has the property of water softening. The developed groundwater system enormously softens the interlayer and leads to strength differences. When the boreholes are inserted into the weak interlayer, the interlayer absorbs more blasting energy than the intact rock, extending the breaking region to the tunnel outline and forming region A. Meanwhile, the stress reflection extends the blasting crack in

region B, resulting in the overbreak in the region. The blasting gas forms air wedges and expands the crack below the interlayer. The process aggravates the overbreak. Additionally, the rapid decrease of the blasting gas pressure and the crack-toward penetrating induce huge energy losses. Hence, the rock is unable to be destroyed. Finally, the falling rate of large bulk increases.

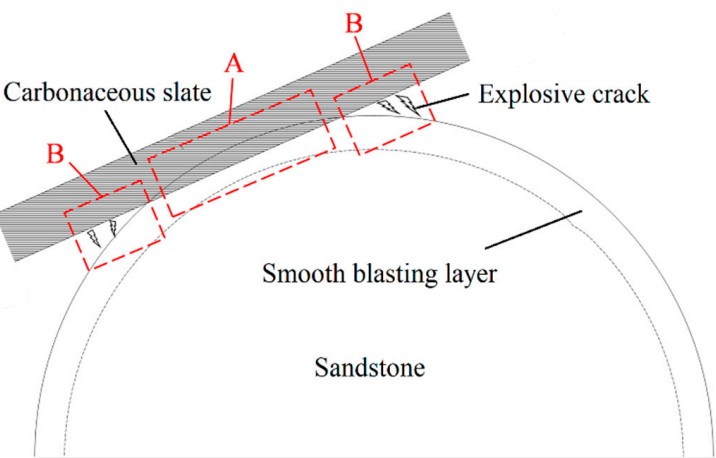

**Figure 22.** Spatial geometry of weak interlayer and crack forming region.

### 3.4.2. Control Measures

For region A, the position of peripheral holes shall be offset close to the excavation outline to reduce the local thickness of the smooth blasting layer. Additionally, the spacing and charging amount of the peripheral holes need to be reduced for the precise blasting control (discrete rock and less damage). For region B, the reflected tensile stress promotes the development of the explosion crack. The guiding holes shall be added in the region of the peripheral hole to guide the crack pattern approaching the excavation outline. Also, the charging amount shall be reduced, too.

The quantitative blasting parameter adjustment is proposed based on the semi-empirical and simulation analysis: For region A, the blasting holes were offset by 10 cm from the origin position, and the hole spacing was adjusted from 50 cm to 40 cm. For region B, guiding holes were inserted between two blasting holes. For the whole overbreak region, the single-hole charge was adjusted from 0.21 kg/m to 0.18 kg/m, and the interval charge structure was applied and the charge was adjusted from the 1 roll to 0.5 rolls.

### 3.4.3. Control Effect

The section of the blasting section is 24 m with 12 cycles. The typical blasting outline and results are shown in Figure 23 and listed in Table 6. The result indicates that the overbreak phenomenon is apparently controlled through design adjusting: the average linear overbreak excavation value decreased from the previous 16.4 cm to 6.1 cm with a 62.5% decrease; the blasting hole residual rate at tunnel walls increased from 19% to 46%; the probability of tunnel underbreak excavation decreased from 33% to 12%. Moreover, the severe block falling and rock losing phenomenon was eliminated, and a significant proportion of reduction of the huge dregs was found. The successful experiment finally promoted the design method and guaranteed the construction safety of the whole tunnel.

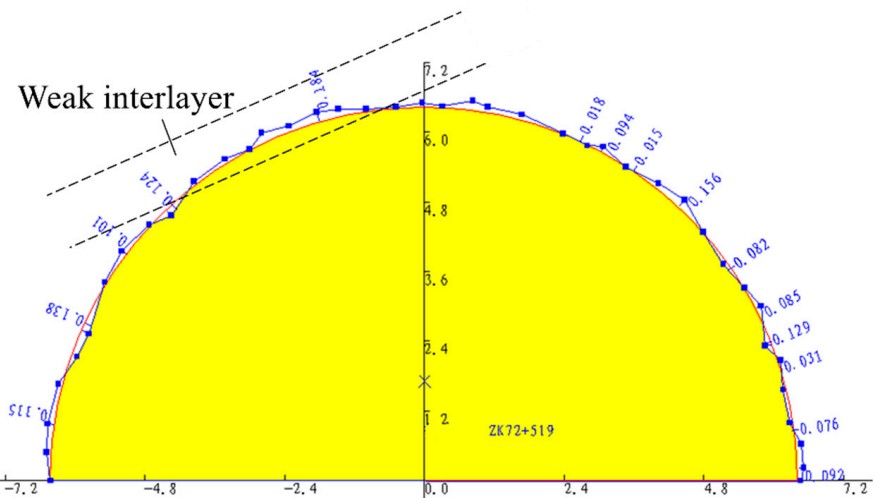

**Figure 23.** Tunnel blasting shaping outline after adjustment of blasting design.

**Table 6.** Comparison of tunnel overbreak and underbreak effects.

| Stage | Statistical Length/m | The Excavation Effect | | | |
|---|---|---|---|---|---|
| | | Average Linear Overbreak/cm | Blast Hole Residue Rate/% | Underbreak Probability/% | Contour Rock Situation |
| Prophase | 8 | 16.4 | 19 | 33 | Difficult in forming arched contours; Rock block loose and dropped heavily. |
| Test | 24 | 6.1 | 46 | 12 | Meeting contour design requirements; No rock block dropping phenomenon. |

## 4. Discussion

The numerical model provides a comprehensive and economic approach for the mechanism analysis at the engineering level. The numerical method's reliability has been validated by related research [36,37]. It is hard to analyze the stress wave propagation via the model test and field monitoring [38]. From the numerical aspect, the weak interlayer leads to a heterogeneous propagation medium, forming a stress barrier at the propagation path. The barrier effect reduces the propagation speed and the peak value of the stress wave but enhances the peak stress in the reflection area; the overall stress wave characteristics are clearly presented. Results from monitoring and model tests only reflect the attenuation of vibration velocity with the propagation path [38]. Hence, the numerical approach has superiority in engineering analysis for practical purposes.

Correspondingly, the stress distribution aggravates the local region damage of the rock mass and relieves the rock damage at blind zones. As a result, the asymmetric damage will increase the difficulty of tunnel blasting design. At the mesoscopic level, theoretical research has revealed the fracture mode near the interlayer; the interlayers' strength and thickness will change the failure mode, the stress concentration, and the regional deformation [39]. Additionally, the propagation in different media forms the transmission angle change, the shear slip of the interlayer–rock interface damage, and the open crack in the interlayer [40].

In numerical modeling, the interface contacts are simplified, which does not consider the interface contact and failure pattern transformation. The simplification reduces the analytical and design precision. Meanwhile, the constitutive model of bilinear kinematic hardening cannot model the large deformation and high strain rate situation. Also, it is hard to reflect the cumulative damage of the rock [37]. The cycle blasting scheme may accumulate on the rock near the outline in practical tunnel blasting engineering. The consideration of the cumulative effect may seriously affect the results.

In summary, the field application indicates that the present precision can be applied to the design, and simplification is acceptable. The blasting design should comprehensively

view the stress wave propagation and rock damage propagation. Notably, the following four aspects should be noted: (1) the aggravated rock damage in the reflection zone; (2) the relieving damage level in the blind zone; (3) the low-strength characteristics of the weak interlayer; and (4) increased destruction under static pressure of explosive gas. The proposed failure modes provide reliable reference on the preliminary judgment of blasting outline form. Hence, more geological parameters are required for design in the actual design procedure, such as the weak interlayer's thickness, distance, and inclination.

Moreover, practical engineering may pass through a geologic body with a complex situation with various interlayer quantities, densities, and mechanical properties. Further research is supposed to be conducted on the multiple effects of the interlayer for design.

### 5. Conclusions

Dynamic blasting models are established to explore the mechanism of the single-hole blasting wave propagation and multi-hole blasting shaping effect with the weak interlayer. Blasting shaping outlines under various spatial geometry are derived from the geometric relation between the interlayer and the excavation outline. Furthermore, the barrier effect of the weak interlayer is discussed; and the inverse blasting design method is proposed according to the predicted blasting outline. The conclusions are listed as follows:

(1) The existence of the weak interlayer in the intact rock has a barrier effect on the stress wave propagation: inducing stress transmission and reflection at the interface between weak and intact rock and causing the redistribution of the stress wave. The geometrical relation of the weak interlayer and blasting hole impacts the barrier effect: the interlayer's thickness and blasting distance promotes the barrier effect, and the blasting energy converges along the vertical direction of the interlayer.

(2) The spatial geometry of the interlayer is classified and generalized, and a simplified conceptual model to analyze the blasting shaping effect is proposed. The tunnel blasting outline is numerically calculated. The interlayer's influence on the blasting shaping effect is discussed from the stress wave propagation and the blasting gas entering. Based on the geometrical relationship, the blasting shaping outlines under various interlayer geometry conditions are derived from providing a reference for the blasting design.

(3) The design strategy considering the barrier effect and the induced uneven rock blasting damage on the blasting design is emphatically discussed. An inverse blasting design optimization utilizing the predicted blasting outline is applied in a test tunnel section. The fitting results validate the effectiveness of the method.

**Author Contributions:** Conceptualization, L.L.; Data curation, M.L., R.H., L.L., N.S. and Y.Z.; Investigation, N.S. and G.Q.; Methodology, M.L. and L.L.; Project administration, R.H.; Resources, G.Q.; Software, R.H.; Supervision, N.S.; Visualization, N.S.; Writing—original draft, M.L.; Writing—review & editing, L.L. All authors have read and agreed to the published version of the manuscript.

**Funding:** Projects funded by the National Natural Science Foundation of China (No. 51978669) are gratefully acknowledged. Besides, thanks to the Support by the Open Sharing Fund for the Large-scale Instruments and Equipments of Central South University (CSUZC202118), and the Science and Technology Project of Guizhou Road and Bridge Group Co., Ltd. (LATJ-12-KY01).

**Informed Consent Statement:** Informed consent was obtained from all subjects involved in the study.

**Data Availability Statement:** Some or all data, models, or code that support the findings of this study are available from the corresponding author upon reasonable request.

**Conflicts of Interest:** The authors declare no conflict of interest.

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
