# Peer review of "Mechanical Mechanism and Shaping Effect of Tunnel Blasting Construction in Rock with Weak Interlayer"

_sustainability, doi:10.3390/su142013278_

Round 1
Reviewer 1 Report
This paper presented interesting numerical works for rock blasting, the dynamic blasting models, for simulating the single hole blasting in rock mass and the drill and blast method for tunnel excavation, are established to investigate the mechanism of blasting wave propagation and tunnel blasting shaping effect with weak interlayers. It is a comprehensive and good work. However, some minor revisions are required to polish the manuscript to make it more concise and precise:
(1) The abstract should summarize the research's objective, main findings, and major conclusions. Since an abstract is frequently given apart from the article, it must be capable of present the stand-out innovations.
(2) In the introduction, literature review is not sufficient, and some references about tunnels are suggested to be added. Such as "https://doi.org/10.1016/j.tust.2021.104074", " https://doi.org/10.1007/s11431-022-2158-9", " https://doi.org/10.1016/j.tust.2022.104735"…
(3) Figures 1-2 have low text quality within the artwork. Please do not re-use the rejected file or try to raise its resolution and re-save it. Because the quality is inadequate to begin with, raising the resolution will not fix the problem. We recommend that you provide us the original format. Rather of embedding photos, we propose replacing figures with vector/editable objects. Eps, ai, tiff, and pdf are the preferred file formats.
(4) Please double-check that all equations are appropriately given. For example, on page 5, line 94, and page 6, line 114.
(5) Line 125: The author should include a reference to the ALE method.
(6) How to determine the numerical modeling interface parameters?
(7) What is the foundation of the original blasting design scheme?
Author Response
Comment 1: The abstract should summarize the research's objective, main findings, and major conclusions. Since an abstract is frequently given apart from the article, it must be capable of present the stand-out innovations.
Response 1: Thank you for the question. The abstract has been re-structured to have mainly the objectives of the study, the main findings and the main conclusions. ( Line 12-23, page 1)
Comment 2: In the introduction, literature review is not sufficient, and some references about tunnels are suggested to be added. Such as "https://doi.org/10.1016/j.tust.2021.104074", " https://doi.org/10.1007/s11431-022-2158-9", " https://doi.org/10.1016/j.tust.2022.104735"…
Response 2: Thank you for the question. In the introduction, we add some references to tunnels.( Line 42, page 1)
Comment 3: Figures 1-2 have low text quality within the artwork. Please do not re-use the rejected file or try to raise its resolution and re-save it. Because the quality is inadequate to begin with, raising the resolution will not fix the problem. We recommend that you provide us the original format. Rather of embedding photos, we propose replacing figures with vector/editable objects. Eps, ai, tiff, and pdf are the preferred file formats.
Response 3: Thank you for the question. We have already replaced the graphics with vector/editable objects. (Line 84, page 3)
Comment 4: Please double-check that all equations are appropriately given. For example, on page 5, line 94, and page 6, line 114.
Response 4: Thank you for the question. We have checked that all the equations are properly given. For example, on page 4, line 97, and page 4, line 116.
Comment 5: Line 125: The author should include a reference to the ALE method.
Response 5: Thank you for the question. Thank you for the question. We have added a reference about ALE algorithm according to the comment. (Line 149, page 6)
Reference :Soutis, C., Mohamed, G., and Hodzic, A. (2011). Modelling the structural response of GLARE panels to blast load. Composite Structures 94(1), 267-276. doi: 10.1016/j.compstruct.2011.06.014
Comment 6: How to determine the numerical modeling interface parameters?
Response 6: Thank you for the question. In this paper, rock and weak inclusions are in co-nodal contact and explosives and air are in co-nodal contact, the model simulates the actual contact relationship between these components by taking a constrained face-to-face contact approach (*CONTACT_TIED_SURFACE_TO_SURFACE). The boundary condition of the model is a reflection-free boundary (*BOUNDARY_NON_REFLECTING).
Comment 7: What is the foundation of the original blasting design scheme?
Response 7: Thank you for your suggestion. The original blasting design scheme was developed by the design institute based on the Safety Regulations for Blasting (GB6722-2014, 2014), Specifications of Excavation Blasting for Hydropower and Water Resources Projects (DL/T 5135-2013, 2013), and the current common tunnel blasting programs..

Reviewer 2 Report
The presented article is devoted to the influence of a weak interlayer on the effects of an explosion. The working scheme of the research is presented in the introduction. Furthermore, the authors present the theoretical part of the research using numerical modeling in the specialized software LS-Dyna, which enables the simulation of dynamic processes. Usually, this program is used, for example, for crash tests. Model input parameters were taken from other articles. Several computational scenarios were assembled for the position and thickness of the weak interlayer. The scenarios were compiled for the independent effect of the charge as well as for the tunnel drilling pattern.
A large number of results are presented in figures and graphs. There are a lot of results from all the scenarios and the orientation in them disappears after a while. An interesting benefit is the practical application around the weak interlayer. Only a small space is devoted to these results in the article. Perhaps the technical details of the blasting could be described more (e.g. the timing of charges).
The practical conclusion was to move the wells away from the weak interlayer to improve the shape of the tunnel excavation. Interesting results were produced by models of a separate charge where the barrier effect of the interlayer was documented.
The drilling pattern of blasting during tunnel excavation is very often modified and changed according to the actual conditions of excavation and the state of the rock, and from that point of view the work does not bring any new knowledge.
Author Response
Response to Reviewer 2 Comments
Dear reviewer#2:
Thank you very much for your comments to our manuscript entitled “Mechanical Mechanism and Shaping Effect of Tunnel Blasting Construction in Rock with Weak Interlayer” (Manuscript ID: 1943013). All the comments are quite insightful and valuable to improve the quality of our manuscript. According to the comments, we have checked the manuscript thoroughly and made a substantial revision. In addition, the revised parts have been highlighted in yellow in this revised manuscript. You can clearly see the changes we made in the revised version that we have uploaded.
Based on your comments, we have revised and edited the language of the article. Meanwhile, we have rewritten the conclusions. Detailed responses to the comments are as follows:
Comment 1: The presented article is devoted to the influence of a weak interlayer on the effects of an explosion. The working scheme of the research is presented in the introduction. Furthermore, the authors present the theoretical part of the research using numerical modeling in the specialized software LS-Dyna, which enables the simulation of dynamic processes. Usually, this program is used, for example, for crash tests. Model input parameters were taken from other articles. Several computational scenarios were assembled for the position and thickness of the weak interlayer. The scenarios were compiled for the independent effect of the charge as well as for the tunnel drilling pattern.
A large number of results are presented in figures and graphs. There are a lot of results from all the scenarios and the orientation in them disappears after a while. An interesting benefit is the practical application around the weak interlayer. Only a small space is devoted to these results in the article. Perhaps the technical details of the blasting could be described more (e.g. the timing of charges).
The practical conclusion was to move the wells away from the weak interlayer to improve the shape of the tunnel excavation. Interesting results were produced by models of a separate charge where the barrier effect of the interlayer was documented.
The drilling pattern of blasting during tunnel excavation is very often modified and changed according to the actual conditions of excavation and the state of the rock, and from that point of view the work does not bring any new knowledge.
Response 1: Thank you for the question. The numerical model provides a comprehensive and economic approach for the mechanism analysis at the engineering level. The numerical method’s reliability has been validated by related research[1,2].
The limitations of the research presented in this paper and the development trends are highlighted. The main elements include.
â‘ In studying the influence of soft and weak inclusions on the effect of tunnel blast forming, the paper only considers the scale and distribution characteristics of individual soft and weak inclusions. In fact, the geological conditions encountered in actual projects are very complex, and the state and mechanical properties of the weak interlayer are highly variable, so more extensive research is needed to consider the effect of multiple weak interlayers on blasting performance.
â‘¡ In engineering practice, tunnel blasting, both in cross-section and longitudinal section, is a "sequential blasting" process, the previous cycle or the previous section of explosive detonation on the subsequent rock body will cause damage effects, and this damage effect in the tunnel cross-section and longitudinal section within a certain range will appear cumulative effect, therefore, how to This is a difficult problem in the field of drilling and blasting tunnels, which is worthy of further study.
- Tian, Z.N.; Zhang, L.W. Numerical analysis of blast wave propagation in rock mass containing weak interlayer. Shenyang Gongye Daxue Xuebao/Journal of Shenyang University of Technology 2010, 32, 349-354.
- Ji, L.; Zhou, C.; Lu, S.; Jiang, N.; Gutierrez, M. Numerical Studies on the Cumulative Damage Effects and Safety Criterion of a Large Cross-section Tunnel Induced by Single and Multiple Full-Scale Blasting. Rock Mech Rock Eng 2021.
Thank you again for giving us the opportunity to strengthen our manuscript with your valuable comments and queries. If you need to contact me, please feel free to contact me by e-mail at mingdfenglei@csu.edu.cn, herui_hfut@163.com, liulinghui888@163.com,
553326623@qq.com, 648395965@qq.com, zhangyunliang88@163.com
Sincerely,
Linghui Liu
School of Civil Engineering
Central South University, China
